# Accelerated HF-rTMS Modifies SERT Availability in the Subgenual Anterior Cingulate Cortex: A Canine [^11^C]DASB Study on the Serotonergic System

**DOI:** 10.3390/jcm11061531

**Published:** 2022-03-10

**Authors:** Yangfeng Xu, Mitchel Kappen, Kathelijne Peremans, Dimitri De Bundel, Ann Van Eeckhaut, Nick Van Laeken, Filip De Vos, Andre Dobbeleir, Jimmy H. Saunders, Chris Baeken

**Affiliations:** 1Department of Psychiatry and Medical Psychology, Ghent Experimental Psychiatry (GHEP) Lab, Ghent University, 9000 Ghent, Belgium; mitchel.kappen@ugent.be (M.K.); chris.baeken@ugent.be (C.B.); 2Department of Veterinary Medical Imaging and Small Animal Othopaedics, Faculty of Veterinary Medicine, Ghent University, 9820 Merelbeke, Belgium; kathelijne.peremans@ugent.be (K.P.); andre.dobbeleir@skynet.be (A.D.); jimmy.saunders@ugent.be (J.H.S.); 3Department of Pharmaceutical Chemistry, Drug Analysis and Drug Information (FASC), Research Group Experimental Pharmacology (EFAR), Center for Neurosciences (C4N), Vrije Universiteit Brussel, 1000 Brussels, Belgium; dimitri.de.bundel@vub.be (D.D.B.); aveeckha@vub.be (A.V.E.); 4Laboratory of Radiopharmacy, Department of Pharmaceutical Analysis, Faculty of Pharmaceutical Sciences, Ghent University, 9000 Ghent, Belgium; nick.vanlaeken@ugent.be (N.V.L.); filipx.devos@ugent.be (F.D.V.); 5Department of Psychiatry, Vrije Universiteit Brussel (VUB), Universitair Ziekenhuis Brussel (UZBrussel), 1000 Brussels, Belgium; 6Department of Electrical Engineering, Eindhoven University of Technology, 5612 Eindhoven, The Netherlands

**Keywords:** canine brain, SERT, dopamine, aHF-rTMS, [^11^C]DASB

## Abstract

Repetitive transcranial magnetic stimulation (rTMS) is thought to partly exert its antidepressant action through the serotonergic system. Accelerated rTMS may have the potential to result in similar but faster onset of clinical improvement compared to the classical daily rTMS protocols, but given that delayed clinical responses have been reported, the neurobiological effects of accelerated paradigms remain to be elucidated including on this neurotransmitter system. This sham-controlled study aimed to evaluate the effects of accelerated high frequency rTMS (aHF-rTMS) over the left frontal cortex on the serotonin transporter (SERT) in healthy beagle dogs. A total of twenty-two dogs were randomly divided into three unequal groups: five active stimulation sessions (five sessions in one day, *n* = 10), 20 active stimulation sessions (five sessions/day for four days, *n* = 8), and 20 sham stimulation sessions (five sessions/day for four days, *n* = 4). The SERT binding index (BI) was obtained at baseline, 24 h post stimulation protocol, one month, and three months post stimulation by a [^11^C]DASB PET scan. It was found that one day of active aHF-rTMS (five sessions) did not result in significant SERT BI changes at any time point. For the 20 sessions of active aHF-rTMS, one month after stimulation the SERT BI attenuated in the sgACC. No significant SERT BI changes were found after 20 sessions of sham aHF-rTMS. A total of four days of active aHF-rTMS modified sgACC SERT BI one month post-stimulation, explaining to some extent the delayed clinical effects of accelerated rTMS paradigms found in human psychopathologies.

## 1. Introduction

The serotonin transporter (SERT) plays a crucial role in human major depression, one of the most common mental illnesses worldwide. Selective serotonin re-uptake inhibitors (SSRIs), the first line pharmacotherapeutic modality for depression, exert their function by modulating the serotonin (5-HT) level through blocking of the serotonin transporter (SERT) [1]. SERT is a transmembrane protein that actively transports 5-HT from the synaptic cleft into the presynaptic neuron, thereby ending the 5-HT neurotransmission. Consequentially, by administering SSRIs, a decreased 5-HT re-uptake speed is acquired, which is accompanied by an improvement of depression [2]. As such, SSRIs mediate a decrease in the SERT density by regulation of SERT expressions [3,4,5] or by direct occupancy, which can account for the long-term clinical effects of this kind of antidepressant treatment [6]. Unsurprisingly, brain regions with the highest density of SERT are involved in the regulation of mood, anxiety, and fear, and they comprise the thalamus, hypothalamus, amygdalae, raphe nuclei, caudate, and putamen [2,7]. In a former study in humans and non-human primates, by measuring the changes of 5-HT1B receptor binding induced by escitalopram, they found the serotonin concentration decreases in serotonergic projection areas after a single, but clinically relevant dose of escitalopram, which may contribute to the understanding of the time-lag between SERT occupancy and the clinical effect of SSRIs [8]. In dogs, areas with high and moderate SERT binding were affected by repeated oral escitalopram administration but cortical binding was not influenced [6]. Another study using single IV-administered vortioxetine and citalopram in non-human primates reported a decrease in SERT binding in the caudate nucleus and putamen [9].

Although SSRIs are considered as the first line antidepressant treatment and are also used for a variety of anxiety disorders, not all patients respond well, and treatment resistance is not uncommon [10,11,12,13,14]. Over the last few decades, other treatment modalities have been introduced to be applied in such cases, including electroconvulsive therapy (ECT), repetitive transcranial magnetic stimulation (rTMS), transcranial direct current stimulation (tDCS), and so on. Rodent research reported increases in SERT availability in the cortex, decreases in SERT mRNA in the raphe nuclei, as well as no effect after ECT [15,16,17,18,19]. On the other hand, for rTMS, a decrease in the SERT transcription was reported in rodents [20]. Using 10-min anodal or cathodal tDCS over left PFC produces a significant acute inhibition of DRN 5-HT neurons, thus resembling the effects exerted by SSRIs, such as fluoxetine, or by PFC DBS in mice [21]. However, extrapolating rTMS results from rodents to humans is not that straightforward. Dogs have proven to be a valid animal model for human neuropsychiatric disorders due to natural occurring psychiatric disease and a larger brain, especially the percentage frontal cortex [22]. Furthermore, nuclear brain imaging studies have demonstrated parallels in alterations in neurotransmitter systems and neuronal function (brain perfusion and metabolism) in humans [23,24,25,26] and dogs [27,28,29] suffering from similar neuropsychiatric disorders. Of interest, regions with high SERT density are also conserved between different species such as rodents [30], pigs [31], cats [32], non-human primates [33], and also dogs [34]. Given that rTMS protocols used in the treatment of human depression can be similarly applied in canine species [35], examining neurobiological effects of novel rTMS protocols in the canine model may add to the fine-tuning of such treatments in humans.

Recently, accelerated (a)rTMS protocols have been introduced, showing similar but faster clinical effects in depressed patients. Here, instead of applying rTMS only once per day, and this during four to six weeks, arTMS protocols deliver the same number of stimuli and sessions spread over only a couple of days [36]. Immediate clinical responses to accelerated aHF-rTMS protocols have been described [37], and delayed clinical effects after accelerated intermittent theta burst stimulation (iTBS) [38], and long-lasting effects of both low- and high-frequency rTMS in mice, in particular on neural plasticity [39,40]; however, the underlying neurobiological mechanisms of rTMS are still largely unclear. Nevertheless, the serotonergic system could be one of the main candidates to moderate the rTMS effects [41].

Therefore, this study aimed to investigate time- and dose-dependent effects of (a)HF-rTMS applied to the left frontal cortex on SERT availability in healthy dogs. First, we hypothesized that [^11^C]-DASB binding alterations occurred in the regions with high SERT density. In line with antidepressant intake, a higher dose resulted in higher occupancy of the SERT binding sites [6], we expected SERT decreases, more pronounced with the active 20 sessions as compared to the 5-session aHF-rTMS protocol. We expected no influences on any of the measurements following sham aHF-rTMS.

## 2. Materials and Methods

### 2.1. Animals

A total of twelve healthy Beagle dogs (5 females neutered, 1 female intact, 5 males neutered, and 1 male intact) were used in this study. For ethical reasons (the local ethical committee only agreed to a limited sample size), 10 of these 12 dogs were randomly selected for reuse. In short, after a three-month washout period (equal to 6 months after the last stimulation session) and a return to baseline SERT BI, the dogs were reused and considered as a new test subject. Hence, 22 dogs (12 used and 10 reused) dogs entered the study. A total of 10 dogs were first allocated to the 5-sessions active group, 8 dogs (6 reused) in the 20-sessions active group, and 4 dogs (4 reused) were in the 5-sessions sham group. The relatively small sample was chosen, and the number of included dogs is in line with our former canine studies with TMS and nuclear brain imaging [35,42]. All dogs are owned by the department of veterinary medical imaging and small animal orthopedics and the department of small animals of the faculty of veterinary medicine. The dogs are permanently housed in groups of 8 on an internal surface of 15 m^2^, with permanent access to an outside area of 15 m^2^. The floor coverings in the inner part consisted of wood shavings. Frequently, toys such as Kongs were given to the animals, and they could run and play outside in an enclosed play area. In addition, students of the faculty of veterinary medicine regularly walked the dogs. This study (EC 2015_38) was approved by the Ghent University Ethical Committee. All animal welfare guidelines imposed by the ethical committee were respected.

### 2.2. Neuronavigation

A frameless neuronavigation system was used to provide the external localization of the left frontal cortex of each dog (stimulation target). First, a tomographical dataset (3T MRI) was acquired; thereafter, neuronavigation was performed as described by Dockx et al. (2017) [43].

### 2.3. aHF-rTMS

The 22 dogs were randomly divided into three unequal groups. The first group consisted of 10 dogs (5 neutered males and females). The second group held 8 dogs (3 neutered females, 1 intact female, 3 neutered males, and 1 chemically neutered male). The last group consisted of 4 animals (2 neutered males, 1 chemically neutered male, and 1 neutered female). Several months prior to the stimulation, positive reinforcement was used to accustom all dogs to the researcher and the experimental room.

All stimulations were applied under general anesthesia as described by Dockx et al. (2017) [42]. Immediately following the induction of anesthesia, the motor threshold of the left motor cortex was determined. A motor threshold (MT) of 100% was defined as the machine output (Magstim Company Limited, Whitland, Wales, UK) that could provoke 5 out of 10 visible muscle contractions in the right upper front limb.

Group 1 received 5 active daily stimulation sessions on the same day, whereas group 2 and 3 received 20 active or sham sessions (5 daily stimulation sessions during consecutive 4 days), respectively. Each stimulation session contained 40 trains of 1.9 s each. The trains were separated by a 12 s intertrain interval (in total 1560 pulses were given per session). The time interval between stimulation sessions was 10 to 15 min. This protocol (20 Hz, 110% MT) was an exact copy of an accelerated HF-rTMS treatment protocol performed in MDD patients at our medical university hospital [44].

### 2.4. Radiosynthesis

N-methylation of the precursor N-desmethyl-DASB (50 μg, ABX, Radeberg, Germany), with a [^11^C]methyl triflate, was performed in order to synthesize the SERT ligand [^11^C]DASB. Radiochemical purity of more than 99% was achieved [34].

### 2.5. Imaging Procedure

All dogs received an intravenous cephalic catheter and were intramuscularly (IM) premedicated with dexmedetomidine 375 μg/m^2^.

All dogs underwent four [^11^C]DASB PET scans: baseline, 24 h post stimulation, 1 month, and 3 months after the last aHF-rTMS treatment session was applied (Figure 1). An [^11^C]DASB bolus was IV injected. A total of thirty minutes after the bolus injections, the dogs were placed in sternal recumbence, with the front limbs extended caudally. A CT scan was taken for attenuation correction. A total of forty minutes after the bolus injection, a 20-min static scan was performed with a PET/CT camera (Gemini PET/CT, Siemens, Eindhoven, The Netherlands).

### 2.6. PET Analysis

Pmod (version 3.405, PMOD Technologies Ltd., Zurich, Switzerland) was used to analyze the PET data. The PET-CT data were fitted onto their corresponding MRI to provide anatomical information. The stereotactic atlas by Dua-sharma et al. (1970) [45] was used to delineate 24 regions (Figure 2) of interest (ROI’s): nucleus caudatus (left and right), hippocampus (left and right), amygdala (left and right), frontal cortex (left and right), pons, medulla, midbrain, temporal cortex (left and right), occipital cortex (left and right), parietal cortex (left and right), thalamus (left and right), anterior cingulate cortex (ACC), posterior cingulate cortex (PCC), subgenual anterior cingulate cortex (sgACC), presubgenual anterior cingulate cortex (presgACC), and cerebellum (excluding the vermis). A binding index (BI), defined as the ratio at equilibrium of specifically bound radioligand to the non-displaceable fraction, was calculated for each ROI at each time point with the cerebellum (excluding the vermis) as a reference region.

### 2.7. Statistical Analysis

Rstudio 1.1.456 (R: A Language and Environment for Statistical Computing; R Core Team; R Foundation for Statistical Computing, Vienna, Austria, 2016, https://www.R-project.org/, accessed on 19 January 2022) with packages MASS (version 7.3-50), doBy (version 4.6-2), sommer (version 3.0), stats (version 3.4.2), and emmeans (version 1.3.0) were used for all linear mixed models. Treatment group and time points were set as fixed-effect factors. The presence of a time point by treatment interaction was also considered in the model. A random intercept was included into the model. The factors time (continuous) and animal (categorical) were set as random factors. The predictor time (t) denotes the different timepoint with t1 the first of three (k − 1 = 4 − 1 = 3) dummies (=1 if time point = “24 h post” or 0 otherwise), t2 the second dummy (=1 if time point = “1 month post” or 0 otherwise), t3 (=1 if time point = “3 months post”, or 0 otherwise). The treatment predictor (T) indicates the different treatment modalities with T1 the first of two (k − 1 = 3 − 1 = 2) dummies (=1 if treatment = “20 sessions active” or 0 otherwise) and T2 (=1 if treatment = “5 sessions active” or 0 otherwise). The reference level (for each region) was set as the concentration at baseline in the control group (intercept). The Welsh–Satterthwaite equation was used to calculate the degrees of freedoms. The type I error was set at 0.05. Normality of the error terms, linearity of the regression function, and homoscedasticity of the error terms were checked using diagnostic plots and statistical tests (Bartlett test of homogeneity of variances and the Shapiro–Wilk normality test).

A first multivariate linear mixed model was fitted. The BI of 23 ROI’s (exclusion of the reference region cerebellum) were set as response variables. A multivariate linear mixed model with heterogeneous unstructured variance was used. The model was written as E(Yt|T1,T2) = β0 + β1t1 + β2t2 + β3t3 + β4T1 + β5T2 + β6t1T1 + β7t2T1+ β8t3T1 + β9t1T2 + β10t2T2 + β11t3T2 with Yt as response variable. The BI of the 23 VOI’s (continuous) was set as response value whereas time and treatment (both categorical) were set as predictor value.

Post hoc, linear contrasts were set up for each ROI, for which the previous model indicated a significant time by treatment interaction. We only performed post-hoc testing if there was a surviving Bonferroni correction for these 23 ROIs.

Any further statistical analysis on significant interaction ROIs was again conducted in R (for detailed version information of the statistical software and packages used, see Appendix A). We fitted the (generalized) linear mixed models (GLMMs) using the ‘lme4’ package [46]. Statistical testing of the model and its factors was performed using ‘car’ packages [46,47]. The sum of squares for the model was estimated using the type III approach, and the statistical significance level was set to *p* < 0.05. Follow-up tests with pairwise comparisons of the estimated marginal means (EMMs) were performed with the ‘emmeans’ package [48] and corrected using the Bonferroni method. All codes and data are made openly available through (https://osf.io/5q7mb/, accessed on 19 January 2022).

## 3. Results

The initial multivariate linear mixed model revealed only a significant time by treatment interaction effect for the sgACC, surviving Bonferroni correction (*p*-value = 0.002) (Table 1).

Based on the multivariate linear mixed model, the post hoc linear contrasts were set up only for the sgACC (Table 2 and Table 3). The linear contrasts revealed a decrease in SERT BI in the sgACC for the 20 active sessions protocol (Table 2: estimate = −0.42; 95% CI (−0.66; −0.18), *p*-value = 0.001; Table 3: estimate = 0.53; 95% CI (0.23; 0.83), *p*-value < 0.001). Of note, after Bonferroni correction, no significant changes in SERT BI were observed for the 20 sham sessions and the five active stimulation sessions protocols.

Given the limited sample size and to verify whether five and 20 active aHF-rTMS differentially affected sgACC SERT BI, we fitted an extra statistical model that best represents this underlying distribution (e.g., normal and gamma), and a series of (G)LMMs were fitted. Only identity link functions were used to increase interpretability of the models and decrease the likelihood of fitting an overly complex model. Based on the Akaike information criterion (AIC), sgACC SERT BI was best represented by a gamma model with an identity-link (AIC = −9.37).

Corresponding model was fit with time (baseline–24 h–1 month–3 months; 4 levels) and treatment group (5 sessions—20 sessions active; 2 levels) as independent variables and subject ID as random intercept. The GLMM showed a significant effect of time, *X*^2^(3, N = 18) = 26.31, *p* < 0.001.

Moreover, a significant interaction effect between time and treatment was found, *X*^2^(3, N = 18) = 9.26, *p* = 0.03. As we are especially interested in this interaction effect, follow-up tests were ran to gain information about the individual contrasts between the two treatment groups at each time moment. By using Bonferroni corrections to control for multiple comparisons, no significant effects were found at an individual moment between the two treatment groups (See Table 4 and Figure 3).

## 4. Discussion

By using stringent statistical analysis, we found SERT BI changes only in the sgACC in healthy beagles. The sgACC comes out as the only significant region of interest, given that in human rTMS research, especially in major depression, the prediction of clinical outcome and clinical improvement can be evaluated by sgACC neurobiological mechanisms [44]. It should be noted that for the current canine study, the sgACC was one of the 23 regions of interest and was not setup as a priori single region to be focused on. Nevertheless, only active aHF-rTMS (20 sessions) over the left frontal cortex decreased SERT BI in sgACC one month after this stimulation protocol, though further investigating with additional statistical models showed that there was no significant difference after follow-up testing with 5 active sessions (1-day). Sham aHFrTMS did not significantly influence SERT BI.

First, this study indicates more prominent decreases in SERT BI when applying 20 active sessions as compared to five active sessions and sham. Although our conservative Bonferroni correction at follow-up did not reveal significance anymore, as is depicted in Figure 3, with of course the notion that these were not human depressed patients but healthy beagles, it supports the assumption that increasing the number of pulses in aHF-rTMS may result in more pronounced serotonergic effects (decreases) and hypothetically may lead to better clinical results [36]. Similar to human research, for the number of sessions applying HF rTMS of the left DLPFC, it has been proved that the clinical effect was higher for a higher number of sessions and rTMS pulses per session [49]. They demonstrated that the responder rate increased significantly when the number of sessions was greater than 10, the total number of pulses delivered per session greater than 1000, and the stimulation intensity greater than 100% resting motor threshold (RMT).

Second, SERT BI decreases were observed one month post stimulation, indicating delayed effects on the serotonergic system. Of interest, the decreases in SERT BI were found in the sgACC, regions known to be highly involved mood regulation. In depressed human patients, successful (non)pharmacological treatment has been associated with a reduction of the hyperactivity of the sgACC [50,51]. Intriguingly, during the last couple of years, the sgACC has been considered as the key region to modulate the clinical effects of (accelerated) rTMS in human depression [52]. Although speculative, the reduction of the SERT BI could be mediated by a decrease in availability of the SERT due to an increase in the extracellular 5-HT levels. Indeed, it has been reported that DASB binding is influenced by endogenous synaptic serotonin [53], and studies evaluating the effect of chronic use of SSRI on the SERT have hinted at an association between a high pre-treatment SERT density in the sgACC and a positive clinical response to the administered SSRIs [54,55]. The mechanisms by which rTMS affects brain activity are still unclear; a leading hypothesis is that these effects are produced through a modulation of neurotransmitter systems including 5-HT [56,57]. There is no acute effect induced by rTMS because it does not directly affect the SERT in the same way as SSRIs do, therefore the delayed effect might be the chronic serotonergic enhancement on plasticity, which still needs further illustration [58]. Based on the results of this study, we conclude that the effects of aHF-rTMS on SERT BI are region- and time-dependent.

These findings could further pave the way to recruit dogs with behavioral disorders, known to be related to the serotonergic dysfunctions, and potentially treat them with aHF-rTMS, or other noninvasive brain stimulation (NIBS) techniques such as transcranial direct current stimulation (tDCS) [59]. At this point, we can only report these serotonin transporter changes without associating this to behavioral effects, because we did not include behavioral assessments here, and the Beagle dogs had no diagnosis of pathological behavior. Of note, this protocol has been applied to anxious and aggressive dogs including different breeds, where in some nearly immediate beneficial effects are observed [60].

Some limitations must be mentioned. Firstly, the aHF-rTMS protocol was only applied in a limited number of healthy dogs, where due to local ethical restrictions, some of them were reused for stimulation. Of note, no significant differences in SERT BI could be observed three months post-stimulation, indicating that in healthy Beagles, even after 20 active aHF-rTMS sessions, serotonergic functioning returns to baseline values. Intriguingly, this might be related to the relapse in clinics, thus maintenance TMS (mTMS) could be mandatory. However, human brain imaging studies of this SERT time pattern are lacking [61]. Future studies should investigate the capacity of mTMS to prevent relapses and evaluate its long-term safety and efficacy, as well as its effects on the serotonergic system [62].A limitation is also that, due to ethical concerns concerning repeated anesthesia, further exploring the immediate effects of both protocols was unfeasible. However, we explored the 24 h, which we considered as an acute effect. Furthermore, there were no differences in baseline measurements, thereby confirming the reuse of the dogs during the experiment with minimal risks of carry-over effects. Nevertheless, the reuse of the same dogs in the analysis compromises generalization. Therefore, our findings should be interpreted cautiously, as we can merely state that our observations only apply to our current study cohort. Secondly, during the sham treatment, an active coil was placed over the left frontal cortex, tilted 90 degrees. An active coil placed in this manner could provoke minor voltages in the underlying cortical tissue [63]. Lastly, the serotonergic aspect is only one of the potential neurotransmitters to be influenced by rTMS protocols, and other neurotransmitter systems were not examined here.

## 5. Conclusions

Only the active 20 session aHF-rTMS protocol resulted in delayed changes in SERT BI, measured with [^11^C]DASB after one month, though significance vanished after stringent post-hoc testing. Nevertheless, our results suggest delayed decreased SERT binding in the sgACC, a key region involved in the therapeutic response of antidepressant therapy in humans and effects. This delayed decrease in SERT BI may point to the presumably therapeutic effect as this is similar to SSRI treatment, where meaningful clinical effects are to be expected after a couple of weeks and not immediately. In addition, these preliminary findings suggest a similar working mechanism of aHF-rTMS, as compared to pharmacotherapy; further research is needed to explore the exact pathways of the (accelerated) rTMS effects on the serotonergic system. Given the similarities between the human and the canine species on the specific anatomical localizations, our observations also indicate that aHF-rTMS treatment may be a valid option to treat dogs diagnosed with mood and anxiety disorders.

## Figures and Tables

**Figure 1 jcm-11-01531-f001:**
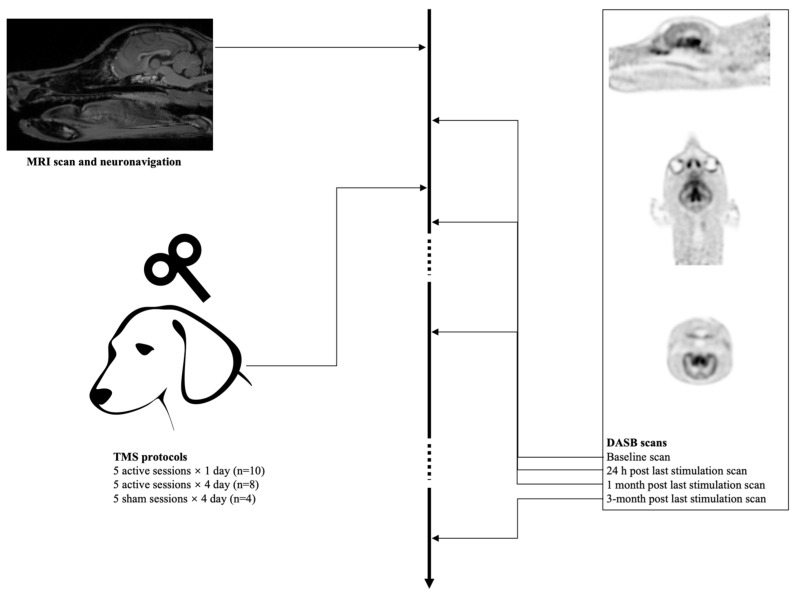
Image procedure.

**Figure 2 jcm-11-01531-f002:**
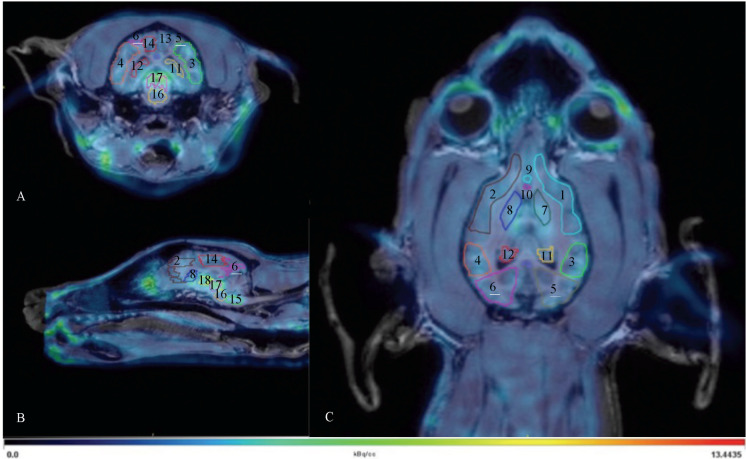
Transversal (**A**), sagittal (**B**), and dorsal (**C**) fusion image ([^11^C]DASB PET scan and MRI). 1: right frontal cortex, 2: left frontal cortex, 3: right temporal cortex, 4: left temporal cortex, 5: right occipital cortex, 6: left occipital cortex, 7: right caudate nucleus, 8: left caudate nucleus, 9: presubgenual anterior cingulate cortex, 10: subgenual anterior cortex, 11: right hippocampus, 12: left hippocampus; 13: right parietal cortex, 14: left parietal cortex, 15: medulla oblongata, 16: pons, 17: midbrain, 18: left thalamus.

**Figure 3 jcm-11-01531-f003:**
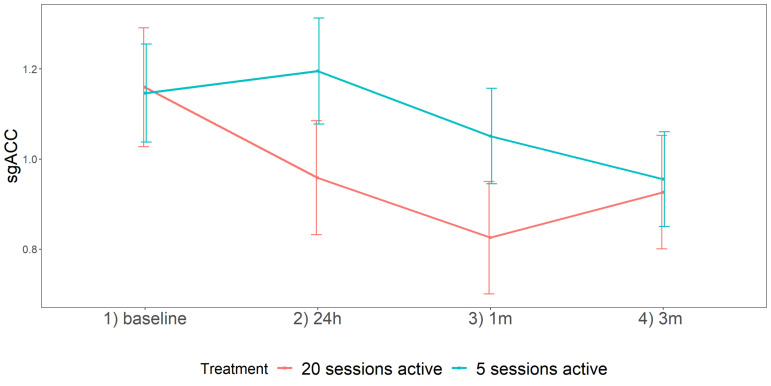
Line plot for sgACC for both 5-sessions and 20-sessions treatment group at each individual time moment. SEs are displayed as error bars.

**Table 1 jcm-11-01531-t001:** Output main effect and interaction test for the predictors time and treatment group.

	Pons	Left Thalamus	Presubgenual Cortex	Subgenual Cortex
Intercept	0.000	0.000	0.000	0.000
24 h	0.297	0.258	0.639	0.580
1 month	0.372	0.631	0.315	0.124
3 months	0.058	0.209	0.064	0.765
20 sessions active	0.826	0.907	0.778	0.144
5 sessions active	0.067	0.462	0.532	0.033 *
24 h: 20 sessions active	0.044 *	0.930	0.352	0.175
1 month: 20 sessions active	0.603	0.036 *	0.015 *	0.002 **
3 months: 20 sessions active	0.294	0.810	0.306	0.367
24 h: 5 sessions active	0.488	0.301	0.817	0.879
1 month: 5 sessions active	0.820	0.809	0.412	0.156
3 months: 5 sessions active	0.570	0.495	0.362	0.316

Note: * *p*-value < 0.05; ** *p*-value < 0.01.

**Table 2 jcm-11-01531-t002:** Multiple comparison for each time point within each treatment group for the sgACC.

		20 Sessions Sham					5 Sessions Active					20 Sessions Active			
	Estimate	SE	LCL	UCL	*p*-Value	Estimate	SE	LCL	UCL	*p*-Value	Estimate	SE	LCL	UCL	*p*-Value
T1-T0	0.01	0.18	−0.27	0.47	0.58	0.13	0.11	−0.09	0.35	0.24	−0.20	0.12	−0.44	0.04	0.10
T2-T0	0.26	0.17	−0.08	0.60	0.13	−0.02	0.11	−0.24	0.20	0.83	−0.42	0.12	−0.66	−0.18	0.001 **
T3-T0	−0.05	0.16	−0.37	0.27	0.77	−0.24	0.10	−0.44	−0.04	0.03	−0.22	0.11	−0.44	0.00	0.05 *
T2-T1	0.16	0.18	−0.21	0.53	0.37	0.16	0.11	−0.38	0.06	0.17	−0.23	0.12	−0.47	0.01	0.06
T3-T1	−0.15	0.17	−0.49	0.19	0.39	−0.37	0.11	−0.59	−0.15	0.002 *	−0.03	0.11	−0.25	0.19	0.80
T3-T2	−0.31	0.16	−0.63	0.01	0.06	−0.22	0.10	−0.42	−0.02	0.04	0.20	0.12	−0.04	0.44	0.09

Note: * *p*-value < 0.05; ** *p*-value < 0.01. LCL = lower control limit, UCL = upper control limit.

**Table 3 jcm-11-01531-t003:** Multiple comparison for each treatment group within each time point for the sgACC.

	5 Sessions Active–20 Sessions Sham					20 Sessions Active–20 Sessions Sham					5 Sessions Active–20 Sessions Active				
	Estimate	SE	LCL	UCL	*p*-Value	Estimate	SE	LCL	UCL	*p*-Value	Estimate	SE	LCL	UCL	*p*-Value
T0	0.42	0.19	0.03	0.81	0.03 *	0.29	0.19	−0.1	0.68	0.15	0.13	0.15	−0.17	0.43	0.41
T1	0.45	0.2	0.04	0.86	0.03 *	−0.01	0.2	−0.42	0.4	0.98	0.46	0.15	0.16	0.76	0.01 *
T2	0.13	0.19	−0.26	0.52	0.48	−0.4	0.19	−0.79	−0.01	0.05 *	0.53	0.15	0.23	0.83	0.00 **
T3	0.22	0.18	−0.15	0.59	0.21	0.11	0.18	−0.26	0.48	0.53	0.11	0.14	−0.17	0.39	0.44

Note: * *p*-value < 0.05; ** *p*-value < 0.01. LCL = lower control limit, UCL = upper control limit.

**Table 4 jcm-11-01531-t004:** Statistics for contrasts between 5-active sessions vs. 20-active sessions treatment at each individual time moment. *p*-values are corrected for multiple comparison using the Bonferroni method.

	b	SE	z	*p*
Baseline	0.0132	0.169	0.078	0.938
24 h	−0.2367	0.171	−1.381	0.167
1 month	−0.2254	0.161	−1.399	0.162
3 months	−0.0288	0.161	−0.178	0.859

## Data Availability

All codes and data are made openly available through (https://osf.io/5q7mb/, accessed on 19 January 2022). The PET data presented in this study are available on request from the corresponding author.

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
