# Peer review of "Accelerated HF-rTMS Modifies SERT Availability in the Subgenual Anterior Cingulate Cortex: A Canine [11C]DASB Study on the Serotonergic System"

_jcm, 2022, doi:10.3390/jcm11061531_

Round 1

Reviewer 1 Report

The Authors addressed all the points and the ms is now strongly improved

Reviewer 2 Report

The authors had addressed most of my previous comments, and they significantly improved the quality of this manuscript. As the authors had made significant modifications to this work, including the title of this article and the main findings resulting from the new analytic method, I regarded this work as a new one to do the review.

This study examined the changes in the binding of the serotonin transporter (SERT) in 22 healthy beagle dogs at baseline and 24 hours, one month, and three months after receiving three kinds of accelerated high-frequency rTMS (aHF-rTMS). The primary aim was to determine the changes in SERT binding with time. The main observations were:

  1. After 20 sessions of active aHF-rTMS, SERT binding in the subgenual anterior cingulate cortex (sgACC) decreased at 1-month post-intervention.
  2. There were no significant effects on SERT binding after receiving five sessions of active aHF-rTMS or 20 sessions of sham aHF-rTMS.

The study is clinically relevant regarding the current unmet needs for treating depression and anxiety by current antidepressants, including the delayed onset of treatment effects (e.g., under adequate doses for 4-6 weeks). Moreover, several time points of PET imaging after intervention provided a valuable opportunity to evaluate the possible alterations that are difficult to perform in human subjects. However, I have the following concerns:

Major comments

  1. Abstract/study aim: In the abstract, the initial part indicated that “Accelerated rTMS may have the potential to result in similar but faster clinical improvement,...”. However, the conclusion part in the abstract also claimed that “...explaining to some extent the delayed clinical effects of accelerated rTMS paradigms found in human psychopathologies.” The main hypothesis of this work is unclear. Is accelerated rTMS paradigms providing faster clinical improvements than convention treatments or not? Is the main observation support their initial hypothesis or not? How to interpret the current results under the framework of the main hypothesis?
  2. Introduction (lines 51-53, lines 99-100, lines 59-63): SSRIs occupy the SERT that also decreased SERT binding revealed by PET imaging. Therefore, it is needed to clarify the claims: is “the decrease in the SERT density” (lines 51-53) originating from the occupancy of SSRIs or the regulation of SERT expressions induced by long-term treatment of SSRIs. It is also warranted to clarify the meaning of “In line with antidepressant intake” in lines 99-100. Similarly, regarding the studies described in lines 59-63, the decreased SERT binding was caused by the occupancy of SSRIs and did not represent the regulation effects.
  3. Introduction (lines 56-59): The interpretations of the NHP study might not follow the results of this study. They measure the changes of 5-HT1B receptor binding induced by escitalopram which did not directly occupy the 5-HT1B receptor but increased 5-HT concentration by blocking SERT.
  4. Results (lines 247-251, figure 3): This study's primary aim is to compare different time effects among varied treatment groups. It might not be meaningful to present the main effect of time on sgACC binding.
  5. Results (lines 255-260, figure 4): As the main control group is the 20 sessions of sham, it is not clear why only the comparison between 5 and 20 sessions of active aHF-rTMS was presented. It would be good also to have the results of 20 sessions of sham in figure 4.
  6. Discussions (lines 281-282): The current results (only the 20 sessions of active aHF-rTMS induced significant changes in SERT binding) did not support the claims.
  7. Discussions (lines 281-308, 2nd & 3rd paragraph in P8): It would be good to reorganize these two paragraphs and focus on the following topics: (1) the importance of the number of active sessions: although this is one of the main points of this study, according to the authors, it lacks relevant discussion. Does the author propose that the aHF-rTMS could provide similar treatment effects and/or biological changes to those of conventional rTMS? If it is the case, it will be interesting to know if the current results compatible with those of previous conventional rTMS studies. (2) Under the framework of time and region-dependent changes of SERT binding, discuss the possible reasons, underlying mechanisms, and clinical implications for the changes in sgACC after one month. These fundamental topics needed to be addressed before the claims for the potential applications of the current imaging protocols.
  8. Discussions (lines 311-312): Although the lack of changes of SERT binding after three months post-treatment could support the re-use of dogs, it also highlights the lack of persistent effects of aHF-rTMS. It is interesting to know if similar patterns occurred in clinical studies and how about the conventional rTMS.

Minor comments

  1. Title: The title “The number of accelerated HF-rTMS sessions modifies SERT…” is a bit strange as “the number” will not modify.
  2. Abstract (lines 25-26): “…in similar but faster clinical improvement,…”, what does such compared to?
  3. Abstract (lines 33-34): The sentence provided the same information as the following sentences.
  4. Introduction (line 64): “…treatment - but also for a…”, might be some typo.
  5. Introduction (line 77): “…especially the percentage frontal cortex [22].”, might be some typo.
  6. Discussions (lines 318-319): “Nevertheless, the reuse of the same dogs in the analysis it compromises generalization.” , might be some typo.
